# Improvement in the Immunity- and Vitamin D_3_-Activity-Related Gene Expression of Coccidiosis-Challenged Ross 708 Broilers in Response to the In Ovo Injection of 25-Hydroxyvitamin D_3_ [note 1]

**DOI:** 10.3390/ani12192517

**Published:** 2022-09-22

**Authors:** Seyed Abolghasem Fatemi, Kenneth S. Macklin, Li Zhang, Ayoub Mousstaaid, Sabin Poudel, Ishab Poudel, Edgar David Peebles

**Affiliations:** 1Department of Poultry Science, Mississippi State University, Mississippi State, MS 39762, USA; 2Department of Poultry Science, College of Agriculture, Auburn University, Auburn, AL 36849, USA

**Keywords:** 25-hydroxyvitamin D_3_, broilers, D_3_ activity-related genes, immunity-related genes, in ovo injection

## Abstract

**Simple Summary:**

Coccidiosis is still considered one of the main diseases affecting the performance and health of poultry reared under intensive production systems. Vitamin D_3_ sources have been shown to reduce the negative effects of a coccidiosis infection with a subsequent improvement in live performance and intestinal immunity. Therefore, the aim of the current research was to explore molecular mechanisms that may play role in an improvement in immunity and vitamin D activity in response to the in ovo injection of vitamin D_3_ (D_3_) and 25-hydroxyvitamin D_3_ (25OHD_3_) alone or in combination in broilers subjected to a coccidiosis infection. In this study, it was shown that the expression of genes linked to an anti-inflammatory response increased and a pro-inflammatory response decreased in the jejunum of 28-day-old broilers (2 weeks after coccidiosis infection) after an in ovo injection of 2.4 μg of 25OHD_3_. Additionally, the in ovo administration of 2.4 μg of 25OHD_3_ increased the expression of genes linked to D_3_ function. In conclusion, the in ovo administration of 2.4 μg of 25OHD_3_ at 18 days of incubation can improve the immunity as well as the D_3_ activity of broilers challenged with coccidiosis.

**Abstract:**

Effects of the in ovo administration of two vitamin D_3_ sources (vitamin D_3_ (D_3_) and 25-hydroxyvitamin D_3_ (25OHD_3_)) on the expression of D_3_ activity- and immunity-related genes in broilers subjected to a coccidiosis infection were investigated. At 18 d of incubation (doi), five in ovo injection treatments were administrated to live embryonated Ross 708 broiler hatching eggs: non-injected (1) and diluent-injected (2) controls, or diluent injection containing 2.4 μg of D_3_ (3) or 2.4 μg of 25OHD_3_ (4), or their combination (5). Birds in the in ovo-injected treatments were challenged at 14 d of age (doa) with a 20× dosage of a live coccidial vaccine. At 14 and 28 doa, the expression of eight immunity-related genes (IL-2, IL-6, IL-10, TLR-4, TLR-15, MyD88, TGF-β4, and IFN-γ) and four D_3_ activity-related genes (1α-hydroxylase, 25-hydroxylase, 24-hydroxylase, and *VDR*) in the jejunum of one bird in each treatment–replicate group were evaluated. No significant treatment effects were observed for any of the genes before challenge. However, at 2 weeks post-challenge, the expression of 1α-hydroxylase, TGF-β4, and IL-10 increased in birds that received 25OHD_3_ alone in comparison to all the other in ovo-injected treatment groups. Additionally, the expression of 24-hydroxylase and IL-6 decreased in birds that received 25OHD_3_ in comparison to those injected with diluent or D_3_ alone. It was concluded that the in ovo injection of 2.4 μg of 25OHD_3_ may improve the intestinal immunity as well as the activity of D_3_ in Ross 708 broilers subjected to a coccidiosis challenge.

## 1. Introduction

Cholecalciferol or vitamin D_3_ (**D_3_**) is a common supplemental source of vitamin D in poultry diets. Vitamin D_3_ is hydroxylated twice to become the functionally active form of vitamin D. Vitamin D_3_ is converted to 25-hydroxycholecalciferol (25OHD_3_) by the action of 25-hydroxylase in the liver and is then hydroxylated by renal cells to 1,25-dihydroxylcholecalciferol [1,25-(OH)_2_D_3_] by 1 α-hydroxylase [1,2]. In the chicken, the conversion of 25OHD_3_ to active form (1,25-(OH)_2_D_3_) is tightly regulated by several factors including vitamin D receptor (***VDR***) activity, 1,25-(OH)_2_D_3_ concentration, and calcium serum levels [3,4]. In comparison to D_3_ at the same level of inclusion, dietary supplementation of 25OHD_3_ more strongly enhances broiler breast meat yield [5,6], protein synthesis rate [6,7], and the density of satellite cells in breast muscle [7]. Additionally, 25OHD_3_ has been shown to more effectively reduce the inflammatory response and increase the humoral immunity of broiler chickens that were challenged with pathogens [8,9]. The expression of 1 α-hydroxylase in the livers of lipopolysaccharide-injected birds increased when they were fed 25OHD_3_, but not when they were fed D_3_ [10]. This may result in increased 1,25-(OH)_2_D_3_ levels in the liver and the suppression of the inflammatory response in 25OHD_3_-fed birds [11]. The expression of 1 α-hydroxylase and *VDR* occurs in the macrophages and in the livers of chickens [12,13]. Therefore, increased biosynthesis of 1,25-(OH)_2_D_3_ from 25OHD_3_ can increase the expression of 1 α-hydroxylase and subsequently increase the expression of *VDR* [13].

In ovo injection techniques in broilers have emerged to allow for the direct administration of particular nutrients or vaccines to embryos, which provides an early stimulation of their immune responses [14]. Additionally, in ovo injection provides a cost-effective and uniform delivery of vaccines with limited contamination issues [15]. Improvements in chicken embryonic and post-hatch development and immune status in response to the in ovo injection of various injected materials, including vitamins, have been observed in previous studies. These observed improvements have been particularly evident in poultry reared under intensive production conditions [16,17]. It is well documented that the in ovo injection of 0.6 to 2.4 μg of 25OHD_3_ can increase broiler breast meat yield [18,19,20] and live performance [18,19,20]. It can also improve the characteristics of hatchling [21], bone quality [22,23], immunity [18,24,25], and small intestine morphology [25] of broilers not challenged with pathogens. Furthermore, the in ovo injection of 2.4 μg of 25OHD_3_ has been shown to increase BWG and breast meat yield [26], and to improve the small intestine morphology and immunity [27] of broilers subjected to a coccidiosis infection. However, the molecular mechanisms by which the in ovo injection of vitamin D_3_ sources affect chicken intestinal histomorphology, muscle formation, or immunity have not been previously investigated. Therefore, the objective of this study was to investigate the impacts of the in ovo injection of D_3_ and 25OHD_3_, alone or in combination on the expression of genes linked to immunity and D_3_ activity in Ross 708 broilers subjected to a coccidiosis infection.

## 2. Materials and Methods

### 2.1. Experimental Design, Egg Incubation, and Coccidial Infection

Fifty Ross 708 broiler hatching eggs were randomly set in each of 5 treatment groups in each of 6 replicate trays (1500 total eggs) in a Chick Master Incubator (Chick Master Incubator Company, Medina, OH, USA). The conditions and arrangement of the treatment-replicate groups of eggs in the incubator were as described by Fatemi et al. [26]. In a previous companion study [26] in which the same incubation regimens were used, no significant treatment effects were found for percentage egg weight loss from 0 to 18 day (**d**) of incubation (**doi**), indicating uniform incubational conditions for all prespecified in ovo injection treatment groups. At 18 doi, a Zoetis Inovoject m (Zoetis Animal Health, Research Triangle Park, NC, USA) multi-egg in ovo injection machine was used to deliver a 50 μL solution volume into each egg. The in ovo injection treatments were: non-injected (1) and diluent injected (2) controls, or diluent injection containing 2.4 μg D_3_ (3) or 2.4 μg 25OHD_3_ (4), or their combination (5). All in ovo injection solutions were freshly prepared in accordance to the procedure described by Fatemi et al. [21,22]. At hatch, chicks belonging to the same treatment group across replicate hatching baskets were pooled prior to placement. A total of 20 birds from each pooled treatment group were randomly placed in each of 2 separate isolation rooms containing wire-floored battery cages (0.76 m × 0.46 m (0.35 m^2^). Four birds were placed in each of 8 replicate cages in each of 5 in ovo injection treatment groups (4 birds × 8 replicates × 5 treatments = 160 total birds in each room). The replicate cages for each of the in ovo injection treatment groups were randomly arranged within each isolation room and room was considered as a blocking factor. A Mississippi State University basal corn-soybean diet as described by Fatemi et al. [18,19,26] was used throughout the 41 d of age (**doa**) study period (Fatemi et al., 2021 [18,19,26]. A coccidial challenge infection was performed at 14 doa according to the method described by Fatemi et al. [26] and Poudel et al. [28,29]. Chicks were left unchallenged in the non-injected treatment group.

### 2.2. Tissue Collection, Total RNA Isolation, Reverse Transcription, and Quantitative Real-Time PCR

At 14 and 18 d of age (**doa**), one bird in each room from each treatment-replicate cage was randomly selected for sampling. Approximately 10 to 20 g of the medial side of each jejunum sample was immediately frozen in liquid nitrogen and stored at −80 °C. Total RNA was isolated using the TRIzol^®^ procedure (Invitrogen, Carlsbad, CA, USA) using 1 mL TRIzol^®^ to every 30 mg of tissue according to the manufacturer’s recommendations. RNA quantification was performed using A NanoDrop 2000 spectrophotometer (Thermo Scientific, Wilmington, NC, USA) and agarose electrophoresis. RNA samples were stored at −80 °C until cDNA synthesis. cDNA synthesis was performed using 200 ng of each total RNA sample according to High Capacity cDNA Reverse Transcription Kit protocol (Applied Biosystems, Foster City, CA, USA) [30]. The resulting cDNA was stored at −20 °C. RT-qPCRreactions were conducted in a 20 μL reaction system containing 10 μL of SYBR Green premix, 1.2 μL of forward primer (10 μM), 1.2 μL of reverse primer (10 μM), 2.0 μL of 5× diluted cDNA, and 5.6 μL of RNase-free water. The program was set at 95 °C for 60 s, followed by 40 cycles of 95 °C for 10 s, and 60 °C for 30 s. Melting curve analysis was performed to analyze the specificity of the primers according to protocol described by Poudel et al. [31]. In total, two endogenous or “housekeeping” genes, plus 20 target genes associated with immune response and D_3_-related activity were identified for RT-qPCR (Table 1). The forward and reverse primers of house-keeping genes (glyceraldehyde-3-phosphate dehydrogenase [*GAPDH*] and 18S ribosomal RNA [*18S*]); genes related to immunity including interleukin (***IL***)*-2*, *IL-6*, *IL-10*, interferon gamma (***IFN-γ***), myeloid differentiation factor 88 (***MyD88***), transforming growth factor beta 4 (***TGF-β4***), Toll-like receptor (***TLR****)-4,* and *TLR-15*; and genes linked to vitamin D activity including 1α-hydroxylase, 24-hydroxylase, 25-hydroxylase, and *VDR* were designed according to the procedure described by Fatemi [31] and Poudel et al. [32]. NormFinder software version 20 was used to normalize the target gene data [33], and the normalization suitability of the expression of *GAPDH* and *18S* genes were tested. The *GAPDH* was an endogenous gene that was most suitable for normalization of the data for the target genes. The fold-change differences were calculated according to the method described by Livak and Schmittgen [34].

### 2.3. Statistical Analysis

Each wire-floored battery cage served as an experimental unit, with all 5 treatments randomly represented in each of 8 replicate cages. A completely randomized experimental design was employed within each room and room served as a blocking factor. A non-injected treatment group was kept in a separate part of the battery cages to eliminate its exposure to coccidial oocysts. Furthermore, a treatment group of birds that have received an in ovo injection solution as well as a coccidial challenge were included. However, in the analysis of effects of coccidial challenge on 28 doa, the non-injected treatment was not included. All house-keeping and target genes were tested for normality using PROC UNIVARIATE and were determined as being normally distributed. Prior to (14 doa) and 2 wk after (28 doa) coccidiosis infection, all gene expression data between and across in ovo injection treatment were analyzed by one-way ANOVA using the procedure for linear mixed models (PROC GLIMMIX) of SAS©, version 9.4 (SAS Institute Inc., Cary, NC, USA). Differences were deemed significant at *p* ≤ 0.05, and Tukey’s least square means comparison was used for means separation [37].

## 3. Results

No significant treatment differences were observed for the immunity- and vitamin D_3_ activity-related genes at 14 doa (Table 2 and Table 3). However, there were significant treatment effects for both genes involved in the immunity and vitamin D3 functions of the broilers at 28 doa (Table 2 and Table 3). The in ovo injection of D_3_ significantly increased the expression of *IL-6* at 28 doa in comparison to the diluent and 25OHD_3_ injection treatment group. Conversely, the expression of *IL-10* was increased in the 25OHD_3_ alone injection treatment group in comparison to that of all other injection treatment groups, and the expression of *IL-10* was higher in the D_3_ + 25OHD_3_ treatment as compared to the diluent-injected and D_3_ alone treatment groups. The expression of *TGF-β4* was also higher in the 25OHD_3_ alone injection treatment when compared to all other injection treatment groups (Table 2). Moreover, the in ovo injection of 25OHD_3_ resulted in a higher expression of 1α-hydroxylase as compared to all other in ovo treatments and resulted in a numerically (*p* = 0.072) lower expression of 24-hydroxylase as compared to the D_3_ + 25OHD_3_ and D_3_ alone treatment groups at 2 wk post infection (Table 3). Across in ovo treatment (Table 4), the expressions of *IL-2*, *IL-6*, *IL-10*, *TGF-β4*, *TLR-4*, 1α-hydroxylase, and *VDR* were significantly higher at 2 wk post-infection (28 doa) in comparison to their expression levels before coccidiosis infection (14 doa).

## 4. Discussion

The objective of this study was to determine the effects of in ovo administration of D_3_ and 25OHD_3_ on the gene expression of broilers subjected to a coccidiosis infection. It is well documented that a coccidial infection generates a chronic intestinal infection as a result of a robust up-regulation of pro-inflammatory cytokines linked to the innate [30,38] or adaptive [8,10,39] immune systems. Among genes that have been reported to be expressed in association with innate immunity responses in chickens infected by coccidiosis are *TLR-4*, *TLR-15*, *MyD88*, and nuclear factor kappa B (***NF-κB***). Zhou et al. [30] reported that broilers subjected to an *E. tenella* infection had a higher level of expression of the *TLR-4* and *TLR15* genes. In addition, layers challenged with *E. tenella* have been shown to exhibit a 2 fold increase in the up-regulation of *TLR15*, *NF-κB*, and *MyD88* gene expression in their ceca between 4 and 24 h post infection [38]. Receptors such as *TLR* are capable of recognizing conserved pathogen-associated molecular patterns [40]. These *TLR* are present in all the developmental stages of the life cycle of various *Eimeria* species [41]. The *MyD88* protein is a downstream adaptor for TLR and is curial for many of the functions of TLRs. Intrinsic to those functions, LR-MyD88 signaling has been shown to trigger inflammatory responses to interior pathogens [42]. In the current study, none of the genes associated with innate immunity were differentially expressed, while the expression of those genes linked to adaptive immunity were altered by in ovo injection treatment. The findings in the current study showed that the in ovo injection of D_3_ alone or in combination with 25OHD_3_ did not alter the expression of genes that were linked to either an immune response or to vitamin D activity. However, in ovo administration of 2.4 μg of 25OHD_3_ alone increased the expression of anti-inflammatory response genes (*IL-10* and *TGF-β4*), and deceased the expression of a pro-inflammatory response gene (IL-6) during the coccidiosis infection. In comparison to D_3_ at the same level of supplemental dietary inclusion, it has likewise been shown that the expression of *IL-10* increased and *IL-1β* decreased in response to 2760 IU/kg of 25OHD_3_ in broilers challenged with coccidiosis [8]. Additionally, dietary 25OHD_3_ at 100 μg/kg increased CD4^+^CD25^+^ cells and deceased CD8^+^CD25^+^ cells in coccidiosis-infected broilers [10] and turkeys [39]. It is well documented that the CD4^+^CD25^+^ cells [37,41], IL-10 [43,44] and *TGF-β* [45] are associated with regulatory T cell (**Tregs**) formation that is facilitated by an up-regulation of the expression of the aforementioned genes. The Tregs are immunity suppression cells that act to inhibit T cell proliferation and cytokine production [46]. Thus, these results indicate that dietary 25OHD_3_ induces an adaptive immune response, and more specifically the activation of an anti-inflammatory response during systemic and chronic infections including coccidiosis. The down-regulation of an inflammatory response in the 25OHD_3_ alone in ovo-injected treatment group could be due to the up-regulation of genes linked to vitamin D_3_ activity. When compared to the diluent and D_3_ alone injected treatment groups in their study, the expression of 1α-hydroxylase increased, and 24-hydroxylase decreased in response to the 25OHD_3_ alone treatment. The conversion of 25OHD_3_ to the active form of vitamin D (1,25-(OH)_2_D_3_) is facilitated by 1α-hydroxylase in the kidney [12], intestine [12], breast and thigh muscles [12], and immune cells including macrophages and T-cells [10]. Moreover, by the action of 24-hydroxylase, 25OHD_3_ can also be converted to the inactive form of vitamin D (24,25-(OH)2D_3_) in many cells in which there are vitamin D receptors. Therefore, an increase in the expression of 1α-hydroxylase or a decrease in the expression of 24-hydroxylase may directly affect vitamin D_3_ activity. In previous companion studies, these current treatments employed were also investigated. In comparison the injection of diluent or D_3_ alone, the in ovo administration of 2.4 μg of 25OHD_3_ improved BW gain from 0 to 28 doa [26], meat yield [26], small intestine morphology [27], and the inflammatory response [27] of broilers subjected to a coccidial infection. Therefore, these morphological and serological improvements observed in previous studies may be due to the stimulation of genes linked to vitamin D activity as well as those eliciting an anti-inflammatory response during a coccidiosis challenge.

Across in ovo treatment, the expression of major vitamin D activity genes including *VDR* and 1α-hydroxylase, that converts 25OHD_3_ to 1,25-(OH)_2_D_3_, was significantly increased in the birds subjected to a coccidial infection. The 25OHD_3_ interacts with *VDR* in various organs and tissues such as the intestines [4,47], muscle cells, bone, kidney, parathyroid gland, pancreas, pituitary, chorioallantoic membrane, and the egg shell gland [48,49]. However, its efficacy is associated with *VDR* and 1α-hydroxylase, that converts 25OHD_3_ to 1,25-(OH)_2_D_3_ [4]. It is well-documented that hydroxylation in the liver is reduced when chickens are subjected to stressful conditions [50], mycotoxicosis [51], and *E-coli* infections [52]. Additionally, hydroxylation in the liver is decreased during severe coccidiosis infections [53]. Thus, these results indicate that the intestinal absorption of D_3_ and its functionality may be reduced during a coccidiosis infection, which may subsequently lead to the additional use of supplemental dietary D_3_ or other D_3_ sources, such as 25OHD_3_ to support of vitamin D function. Furthermore, the expression of *IL-2*, *IL-6*, and *TLR-4* involved in pro-inflammatory responses significantly increased during the coccidiosis infection in this study. Several studies have reported the elevation in the expression of *IL-2* and *IL-6* during various avian *Emeria* infections [54,55,56]. Likewise, *TLR-4* categorized as a pathogenic infection indicator, has been shown to be up-regulated during an *E. tenella* infection [30]. Therefore, the broilers in the current study that experienced a successful *Emeria* infection exhibited a measurable immune reaction.

## 5. Conclusions

In conclusion, the aim in the current research was to identify the genes that are linked to an immune response and to vitamin D activity in the jejunum of the small intestine of broilers subjected to an *Emeria* infection after having received individual or combinational in ovo injection of 2 vitamin D_3_ sources. Beside upon the findings in this study, it is suggested that the coccidiosis infection had significant effects on the expression of genes involved in vitamin D, function (1α-hydroxylase and *VDR*), and a pro-inflammatory response (*IL-2*, *IL-6* and *TLR-4*). Furthermore, the in ovo administration of 2.4 μL of 25OHD_3_ resulted in the up-regulation of genes linked to an anti-inflammatory response (*IL-10* and *TGF-β4*). Moreover, the expression of 1α-hydroxylase increased and 24-hydroxylase deceased in response to the in ovo injection of 2.4 μL of 25OHD_3_. However, the other in ovo-injection treatments investigated in this study did not display significant effects on the expression of genes associated with either an inflammatory reaction or to vitamin D function. These results, therefore, demonstrate that the amniotic in ovo administration of 25OHD_3_ at 18 doi may more quickly ameliorate the negative effects of a coccidiosis infection by means of its effects on intestinal vitamin D activity or an inflammatory reaction. Further research is required to determine the regulatory effects of the in ovo administration of different vitamin D_3_ sources on the immune response and intestinal development of broilers subjected a coccidiosis infection.

## Figures and Tables

**Table 1 animals-12-02517-t001:** Real-time PCR primers and GenBank accession numbers of chicken housekeeping genes, and target genes associated with immunity and vitamin D-related activity.

Gene Symbol	Accession No	Type	Orientation	Sequence (5′ to 3′)	Length (nt)	Amplicon Size (bp)	Reference
*RNA 18S*	M59389.1	Housekeeping	Forward	GCCAACAGAGAGAAGATGACAC	22	140	-
			Reverse	GTAACACCATCACCAGAGTCCA	22		
*GAPDH*	NM204305	Housekeeping	Forward	GTAAACCATGTAGTTCAGATCGATGA	26	72	-
			Reverse	GCCGTCCTCTCTGGCAAAG	19		
*IL-2*	AY386204	Immunity-related	Forward	AGTCTTACAGGTCTAAATCACACC	24	102	-
			Reverse	CACAAAGTTGGTCAGTTCATGG	22		
*IL-6*	AB559572	Immunity-related	Forward	GCGAGAACAGCATGGAGATG	20	143	Al-Zghoul et al. [35]
			Reverse	GTAGGTCTGAAAGGCGAACAG	21		
*IL-10*	AJ621614	Immunity-related	Forward	AGCAGATCAAGGAGACGTTC	20	103	-
			Reverse	ATCAGCAGGTACTCCTCGAT	20		
*IFN-γ*	AJ001678	Immunity-related	Forward	GTGAAGAAGGTGAAAGATATCATGGA	26	71	-
			Reverse	GCTTTGCGCTGGATTCTCA	19		
*TGF-β4*	M31160.1	Immunity-related	Forward	GGGGTCTTCAAGCTGAGCGT	20	119	Brisbin et al. [36]
			Reverse	TTGGCAATGCTCTGCATGTC	20		
*TLR-4*	AY064697	Immunity-related	Forward	AGTCTGAAATTGCTGAGCTCAAAT	24	190	-
			Reverse	GCGACGTTAAGCCATGGAAG	20		
*TLR-15*	NM_001037835	Immunity-related	Forward	GGCTGTGGTATGTGAGAATG	20	113	-
			Reverse	ATCGTGCTCGCTGTATGA	18		
*MyD88*	NM_001030962	Immunity-related	Forward	ATGGGCATGGAACAGAGATG	20	138	-
			Reverse	GCAAGACATCCCGATCAAAC	20		
1α-hydroxylase	XM_422077	Vitamin D activity-related	Forward	TCGTGGCAGGAATACAGAGA	20	125	-
			Reverse	ACTGCCACATCTTTGGGTTT	20		
25-hydroxylase	NM_001277354	Vitamin D activity-related	Forward	GCTGTCACTGGGATTCTTTGC	21	160	Shanmugasundaram and selvaraj [12]
			Reverse	CCAACCGAAAGGCACAAGTC	20		
24-hydroxylase	AF019142.1	Vitamin D activity-related	Forward	AAACCCTGGAAAGCCTATCG	20	133	Shanmugasundaram and selvaraj [12]
			Reverse	CCAGTTTCACCACCTCCTTG	20		
*VDR*	AF011356.1	Vitamin D activity-related	Forward	CGTGAGAAGCAAATTCAGCA	20	157	-
			Reverse	GAGGTCCAGGTTGGAAAACA	20		

**Table 2 animals-12-02517-t002:** Effects of in ovo injection treatment (non-injected, diluent-injected (50 μL), and 50 μL of diluent containing 2.4 μg of vitamin D_3_ (**D_3_**), 2.4 μg of 25-hydroxycholecalciferol (**25OHD_3_**), or 2.4 μg of D_3_ and 2.4 μg of 25OHD_3_ (**D_3_ + 25OHD_3_**) on the fold change expression of immune-related genes at 14 and 28 d of age (doa).

Treatment	n	IL-2	IL-6	IL10	IFN-γ	TGF-β4	TLR4	TLR-15	MyD88
14 doa									
Non-injected ^1^	8	0.90	1.53	1.14	0.87	0.91	0.93	1.53	1.10
Diluent ^2^	8	1.04	1.07	1.04	1.06	1.06	1.01	1.12	1.03
D_3_ ^3^	8	1.12	1.69	1.27	1.49	1.03	1.20	1.38	1.31
25OHD_3_ ^4^	8	1.05	1.06	1.33	0.91	2.16	0.92	0.95	1.17
D_3_ + 25OHD_3_ ^5^	8	1.00	1.03	1.11	0.80	1.14	1.11	1.01	1.11
SEM		0.195	0.435	0.309	0.196	0.522	0.178	0.277	0.153
*p*-value		0.758	0.489	0.869	0.103	0.274	0.522	0.556	0.763
28 doa									
Diluent	8	1.05	1.06 ^b^	1.18 ^c^	1.10	1.03 ^b^	1.17	1.06	1.04
D_3_	8	1.25	4.41 ^a^	1.70 ^c^	0.89	1.53 ^b^	1.34	0.96	1.31
25OHD_3_	8	1.41	0.93 ^b^	9.00 ^a^	0.46	4.70 ^a^	1.56	0.97	1.34
D_3_ + 25OHD_3_	8	1.50	2.52 ^ab^	3.40 ^b^	1.03	2.29 ^b^	1.47	1.50	1.40
SEM		0.212	0.832	0.553	0.223	0.505	0.279	0.223	0.218
*p*-value		0.471	0.012	0.001	0.198	<0.001	0.777	0.294	0.660

^a–c^ Treatment means within the same column within effect with no common superscripts are significantly different (*p* ≤ 0.05). ^1^ Eggs that were not injected with any solution and also were not challenged with coccidiosis vaccine at 14 doa. ^2^ Eggs injected with 50 μL of commercial diluent at d 18 of incubation and that were not challenged with coccidiosis vaccine at 14 doa. ^3^ Eggs injected with 50 μL of commercial diluent containing vitamin 2.4 μg of D_3_ at d 18 of incubation and that were also challenged with coccidiosis vaccine at 14 doa. ^4^ Eggs injected with 50 μL of commercial diluent containing 2.4 μg of 25OHD_3_ at d 18 of incubation and that were also challenged with coccidiosis vaccine at 14 doa. ^5^ Eggs injected with 50 μL of commercial diluent containing 2.4 μg of D_3_ and 2.4 μg of 25OHD_3_ at d 18 of incubation and also were challenged with coccidiosis vaccine at 14 doa.

**Table 3 animals-12-02517-t003:** Effects of in ovo injection treatment (non-injected, diluent-injected (50 μL), and 50 μL of diluent containing 2.4 μg of vitamin D_3_ (**D_3_**), 2.4 μg of 25-hydroxycholecalciferol (**25OHD_3_**), or 2.4 μg of D_3_ and 2.4 μg of 25OHD_3_ (**D_3_ + 25OHD_3_**) on the fold change expression of vitamin D activity-related genes at 14 and 28 d of age (**doa**).

Treatment	n	1α-hydroxylase	25-hydroxylase	24-hydroxylase	*VDR* ^1^
14 doa					
Non-injected ^2^	8	1.01	0.89	2.02	0.91
Diluent ^3^	8	1.02	1.02	1.14	1.01
D_3_ ^4^	8	1.34	1.17	2.64	1.32
25OHD_3_ ^5^	8	1.34	1.07	1.49	1.19
D_3_ + 25OHD_3_ ^6^	8	1.16	1.03	1.89	1.15
SEM		0.246	0.160	0.606	0.195
*p*-value		0.549	0.580	0.299	0.381
28 doa					
Diluent	8	1.04 ^b^	1.02	1.03	1.25
D_3_	8	1.16 ^b^	1.13	2.08	1.82
25OHD_3_	8	3.43 ^a^	1.25	0.47	1.56
D_3_ + 25OHD_3_	8	1.80 ^b^	1.91	2.08	1.70
SEM		0.383	0.329	0.437	0.274
*p*-value		0.001	0.243	0.072	0.512

^a, b^ Treatment means within the same column within effect with no common superscripts are significantly different (*p* ≤ 0.05). ^1^ Vitamin D receptor. ^2^ Eggs that were not injected with any solution and also were not challenged with coccidiosis vaccine at 14 doa. ^3^ Eggs injected with 50 μL of commercial diluent at d 18 of incubation and that were not challenged with coccidiosis vaccine at 14 doa. ^4^ Eggs injected with 50 μL of commercial diluent containing vitamin 2.4 μg of D_3_ at d 18 of incubation and that were also challenged with coccidiosis vaccine at 14 doa. ^5^ Eggs injected with 50 μL of commercial diluent containing 2.4 μg of 25OHD_3_ at d 18 of incubation and that were also challenged with coccidiosis vaccine at 14 doa. ^6^ Eggs injected with 50 μL of commercial diluent containing 2.4 μg of D_3_ and 2.4 μg of 25OHD_3_ at d 18 of incubation and also were challenged with coccidiosis vaccine at 14 doa.

**Table 4 animals-12-02517-t004:** Effects of coccidiosis infection on the fold change expression of genes linked to immunity and vitamin D activity before and after coccidiosis infection.

Genes	n	14 doa ^1^	28 doa ^2^	SEM	*p*-Value
*IL-2*	16	1.017 ^b^	1.30 ^a^	0.130	0.031
*IL-6*	16	1.34 ^b^	2.48 ^a^	0.508	0.028
*IL10*	16	1.21 ^b^	3.82 ^a^	0.581	0.002
*IFN-γ*	16	1.08	0.88	0.139	0.135
*TGF-β4*	16	1.29 ^b^	2.39 ^a^	0.410	0.010
*TLR4*	16	1.02 ^b^	1.39 ^a^	0.156	0.021
*TLR-15*	16	1.25	1.12	0.185	0.509
*MyD88*	16	1.15	1.27	0.124	0.380
1α-hydroxylase	16	1.18 ^b^	1.86 ^a^	0.260	0.011
25-hydroxylase	16	1.04	1.33	0.177	0.110
24-hydroxylase	16	1.82	1.42	0.377	0.285
*VDR*	16	1.11 ^b^	1.58 ^a^	0.153	0.003

^a, b^ Treatment means within the same column within effect with no common superscripts are significantly different (*p* ≤ 0.05). ^1^ 14 d of age, when all birds were unchallenged. ^2^ 28 d of age (14 days post-infection), when birds were challenged with coccidiosis vaccine.

## Data Availability

None of the data were deposited in an official repository.

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
