# Peer review of "Improvement in the Immunity- and Vitamin D_3_-Activity-Related Gene Expression of Coccidiosis-Challenged Ross 708 Broilers in Response to the In Ovo Injection of 25-Hydroxyvitamin D_3_ [Author-notes fn1-animals-12-02517]"

_animals, 2022, doi:10.3390/ani12192517_

Round 1
Reviewer 1 Report
1. Do authors use a different method to prove the RT-qPCR results about the immunity- and D3 activity-related gene expression, like ELISA or other assays? It’s better to add them to the manuscript.
2. Line 121, change “form” to “from”
3. Line215, add “14 days post-infection” to 28 d of age
Author Response
Reviewer1:
Do authors use a different method to prove the RT-qPCR results about the immunity- and D3 activity-related gene expression, like ELISA or other assays? It’s better to add them to the manuscript.
Answer: Thank you for the suggestion, to address the suggestion, the current manuscript is companion to a previous study in which we previously reported the immunological effects of the same in ovo injection vitamin D3 treatments when birds were a challenged with a coccidiosis infection. These results are also presented in the current draft from lines 214-218
“The current treatments employed were also investigated in the previous companion studies. In comparison to the injection of diluent or D3 alone, the in ovo administration of 2.4 μg of 25OHD3 improved BW gain from 0 to 28 doa [27], meat yield [27], small intestine morphology [28], and the inflammatory response [28] of broilers subjected to a coccidial infection”. However, this paper is only focused Gene expression and no ELIZA or other immunological variables were presented in this paper. The companion study cited as reference [28], and the serum “Nitric oxide” and “Interleukin 1β (IL-1β)” were measured before and after coccidiosis challenge.
- Line 121, change “form” to “from”
Answer: The relevant correction was performed in the text
- Line 215, add “14 days post-infection” to 28 d of age
Answer: The relevant correction was applied in the text

Reviewer 2 Report
I find this manuscript not suitable for publication because although there is a coccidiosis challenge and researchers provided vitamin D3 to alleviate effects, we do not know anything about the outcome of coccidiosis. I mean, for example that Vit3 may have a positive antiinflammatory effect but chickens to die one day after the day of measurement. For this, reason I ma sorry to decline publication, without performance results.
Author Response
Reviewer2:
I find this manuscript not suitable for publication because although there is a coccidiosis challenge and researchers provided vitamin D3 to alleviate effects, we do not know anything about the outcome of coccidiosis. I mean, for example that Vit3 may have a positive antiinflammatory effect but chickens to die one day after the day of measurement. For this, reason I ma sorry to decline publication, without performance results.
Answer:
The reviewer`s comments are well taken. However, it is necessary to state that the current paper is the third part of companion studies. In the first study, the focus was on live performance, meat yield and oocyst counts, and the second study was related to changes in the immunology and small intestine morphology of broilers challenged with coccidiosis and that received in ovo injections of various forms of vitamin D3. In the previous published studies, the same treatments were used as in this study. In order to confirm the aforementioned information, the summary of the results of the previous companion publications (references 27 and 28) are presented in the introduction and discussion section to properly address previous related findings.
Introduction (Lines 82-85):
“Furthermore, the in ovo injection of 2.4 μg of 25OHD3 has been shown to increase BWG and breast meat yield [27], and to improve the small intestine morphology and immunity [28] of broilers subjected to a coccidiosis infection”.
Discussion:
Lines (214-218) “The current treatments employed were also investigated in the previous companion studies. In comparison to the injection of diluent or D3 alone, the in ovo administration of 2.4 μg of 25OHD3 improved BW gain from 0 to 28 doa [27], meat yield [27], small intestine morphology [28], and the inflammatory response [28] of broilers subjected to a coccidial infection”.
First reference [27]
- Fatemi, S.A.; Elliott, K.E.C.; Bello, A.; Peebles, E.D. Effects of the in ovo injection of vitamin D3 and 25-hydroxyvitamin D3 in Ross 708 broilers subsequently challenged with coccidiosis. I. performance, meat yield and intestinal lesion. Poult. Sci. 2021d, 100, 101382. https://doi.org/10.1016/j.psj.2021.10138.
Second reference [28]
- Fatemi, S.A.; Elliott, K.E.C.; Bello, A.; Macklin, K.S.; Peebles, E.D. Effects of the in ovo injection of vitamin D3 and 25-hydroxyvitamin D3 in Ross 708 broilers subsequently challenged with coccidiosis: II. Immunological and inflammatory responses and small intestine histomorphology. Animals (Basel). 2022, 12, 1027. doi: https://doi.org/10.3390/ani12081027.
Therefore, authors unanimously disagree with the reviewer comments that the performance data are not presented. Furthermore, in the Materials and Methods section, it was not mentioned that all birds in the experiment were killed before or after coccidiosis infection. The sampling days were 14 and 28 d and the study lasted through 41 d. This information was presented in the current draft on Lines 115-117
A Mississippi State University basal corn-soybean diet as described by Fatemi et al. [19,20,27] was used throughout the 41 d of age (doa) study period (Fatemi et al., 2021 [19,20,27].
Also, there is a clear statement that only 1 bird per cage was used for sampling and in each cage, 4 chicks were placed at hatch as indicated on Lines 122-123 and Lines 113-114.
Lines 111-113:
Four birds were placed in each of 8 replicate cages in each of 5 in ovo injection treatment groups (4 birds x 8 replicates x 5 treatments =160 total birds in each room).
Line 122-123:
“At 14 and 18 d of age (doa), one bird from each treatment-replicate cage in each room was randomly selected for sampling”.

Reviewer 3 Report
Dear authors
The manuscript "Improvement in the immunity- and vitamin D3..." is well written and on a topic relevant to the field.
The introduction gives an overview of the topic to be investigated.
The results are presented objectively and are compatible with the methodologies used.
The discussion is well developed and compatible with the results obtained.
Author Response
Reviewer 3:
The manuscript "Improvement in the immunity- and vitamin D3..." is well written and on a topic relevant to the field.
The introduction gives an overview of the topic to be investigated.
The results are presented objectively and are compatible with the methodologies used.
The discussion is well developed and compatible with the results obtained.
Answer:
Thank you so much for the comments.

Round 2
Reviewer 2 Report
Authors have provided adequate data. Revision is satisfactory.